# Effects of Pre-Treatment Using Plasma on the Antibacterial Activity of Mushroom Surfaces

**DOI:** 10.3390/foods10081888

**Published:** 2021-08-15

**Authors:** Sarmistha Mitra, Mayura Veerana, Eun-Ha Choi, Gyungsoon Park

**Affiliations:** 1Plasma Bioscience Research Center and Department of Plasma-Bio and Display, Kwangwoon University, Seoul 01897, Korea; sarmisthacu@gmail.com (S.M.); mayuraveerana@gmail.com (M.V.); ehchoi@kw.ac.kr (E.-H.C.); 2Department of Electrical and Biological Physics, Kwangwoon University, Seoul 01897, Korea

**Keywords:** mushroom, non-thermal atmospheric pressure plasma, antimicrobial quality, food sanitation, plasma-treated water

## Abstract

Although non-thermal atmospheric pressure plasma is an efficient tool for preventing post-harvest microbial contamination, many studies have focused on the post-treatment of infected or contaminated foods. In this study, we examined the antimicrobial quality of mushrooms pre-treated with a non-thermal atmospheric pressure plasma jet (NTAPPJ) or plasma-treated water (PTW). The CFU (Colony Forming Unit) number of *Escherichia coli* inoculated on surfaces of mushrooms pre-treated with NTAPPJ or PTW was significantly reduced (about 60–75% for NTAPPJ and about 35–85% for PTW), and the reduction rate was proportional to the treatment time. Bacterial attachment and viability of the attached bacteria were decreased on NTAPPJ-treated mushroom surfaces. This may be caused by the increased hydrophilicity and oxidizing capacity observed on NTAPPJ-treated mushroom surfaces. In PTW-treated mushrooms, bacterial attachment was not significantly changed, but death and lipid peroxidation of the attached bacteria were significantly increased. Analysis of mushroom quality showed that loss of water content was greater in mushrooms treated with NTAPPJ compared to that in those with no treatment (control) and PTW treatment during storage. Our results suggest that pre-treatment with NTAPPJ or PTW can improve the antibacterial quality of mushroom surfaces by decreasing bacterial attachment (for NTAPPJ) and increasing bacterial lipid peroxidation (for both NTAPPJ and PTW).

## 1. Introduction

A healthy diet, containing low non-digestible carbohydrates and dietary fiber, has a high nutritional value for humans [1,2]. Foods such as beans, legumes, bran cereal, mushrooms, and some other vegetables and fruits provide a good source of fiber, and their consumption is reported to reduce cardiovascular disease risk, enhanced laxation, satiety, and decrease postprandial blood glucose [2,3]. Mushrooms are considered as functional foods having high nutritional value, in which both low-digestible and non-digestible carbohydrates, such as β-glucans, raffinose, chitin, resistant starch, and oligosaccharides, are abundant [2,4].

*Flammulina velutipes* is the common and edible mushroom particularly popular in Asia [5]. Studies have shown that the mushroom possesses various pharmacological properties, including anticancer [6], antimicrobial [7], anti-inflammatory [8], antioxidant [9], and immunomodulatory properties [9], and reduces blood cholesterol [10]. Mushrooms are cultivated widely and are available throughout the north-temperate regions, including North America, Europe, and Asia, because of their desirable taste, aroma, and high nutritional value. Harvested mushrooms are metabolically active, undergo ripening and senescence processes, and are susceptible to contamination and microbial growth that must be controlled to ensure post-harvest quality. Remarkably, fresh mushrooms are eaten raw or after minimal processing, and thus food pathogen contamination can enhance the risk of outbreaks of foodborne-pathogen-related infections [11], caused by *Listeria monocytogenes* [12], *Salmonella enteritidis* [13], and *Escherichia coli* O157:H7 [14]. When spoilage and pathogenic microorganisms come in contact with fresh products, they can rapidly and strongly adhere themselves to surfaces, eventually forming biofilms on product surfaces [15,16]. Furthermore, pathogenic organisms also increase the degradation rate of the mushroom, as microbial contamination is the most common reason of food spoilage. Notably, protecting degradable food from microbial contamination in the post-harvest stage is one of the greatest challenges for long-term preservation and quality maintenance.

Post-harvest antimicrobial technologies are widely used to maintain food quality and include temperature management and various physical (heat, irradiation, and edible coatings), chemical, and gaseous (nitric oxide, sulfur oxide, ozone, ethylene) treatments [17]. As a potential tool, non-thermal atmospheric pressure plasma has recently received attention in the area of post-harvest sanitation and disinfection because enormous amounts of data show that plasma is effective in reducing microbial contamination on fresh products and foods [18,19,20,21,22,23,24]. Numerous studies have shown that plasma or plasma-treated liquids are very effective in the sanitation and extension of shelf-life of post-harvested fruits and vegetables [19,21,22,25]. In addition, plasma can decontaminate and reduce microbial attacks on packaged fresh products [26]. Although rarely explored, efficient sanitation and extension of the shelf-life of post-harvested mushrooms during storage were observed after mushrooms were soaked in plasma-treated water in a current study [27].

Fruits and vegetables have been main targets for post-harvest quality control using plasma in many studies, and therefore, data from other post-harvest products need to be accumulated for better understanding the antimicrobial activity of plasma. In addition, many studies focus on the effect of post-treatment with plasma on the sanitation of fresh produce harboring natural microbiota or artificially inoculated microorganisms [20]. Antimicrobial effects of pre-treatment with plasma (priming treatment using plasma) are rarely explored on post-harvest living products. Effects of pre-treatment with plasma on antimicrobial activity have been well demonstrated in non-living materials. Surfaces of materials are modified to be antimicrobial after treatment with plasma [28,29,30], and plasma can facilitate surface grafting with antimicrobial compounds [31]. Priming treatment of fresh produce using plasma can be an effective strategy for developing more reliable technologies for post-harvest quality control. Using Enoki mushroom (*F. velutipes*) as a target post-harvested product, we investigated the effects of priming treatment using plasma (non-thermal atmospheric pressure plasma jet; NTAPPJ or plasma-treated water; PTW) on antimicrobial quality of mushrooms and the underlying mechanisms in this study.

## 2. Materials and Methods

### 2.1. Sample Preparation

Enoki mushrooms (*Flammulina velutipes*) were purchased in a vacuum-sealed package from a local market (Seoul, Korea) for this study and visually checked to avoid damage and contamination. The purchased mushroom was identified by expert personnel. The mushrooms were cut into 10 pieces, 2 cm long, for plasma treatment. *Escherichia coli* K12 (KCTC1116) was graciously provided by the Korean Collection for Type Cultures in Korea Research Institute of Bioscience and Biotechnology (Jeongeup-si, Jeollabuk-do, Korea) and used to inoculate the mushroom surfaces. *Escherichia coli* K12 was maintained on TSA (tryptic soy agar; MB Cell, KisanBio, Seoul, Korea) and propagated in TSB (tryptic soy broth; MB Cell). One or two colonies from the TSA culture plate were suspended in 1 mL of TSB, and then 10 µL of the suspension was inoculated into 15 mL of TSB. The inoculated tube was incubated at 37 °C with shaking overnight (16–20 h). Bacterial cells were then pelleted using centrifugation at 3134× *g* for 10 min. The bacterial cell pellet was washed with saline (0.9% NaCl solution) and then resuspended in new saline. The optical density of the bacterial suspension was adjusted to approximately 0.2 at 600 nm. This suspension was used for inoculation.

### 2.2. Plasma Device and Preparation of Plasma-Treated Water

An NTAPPJ device (Figure 1a) from the Plasma Bioscience Research Center in Kwangwoon University (Seoul, Korea) was used in the treatment of mushroom pieces (Figure 1b) and deionized water (Figure 1c). The structure of the plasma device, physical properties, and OES (optical emission spectroscopy) analysis of the plasma jet were described in a previous study [32]. Plasma was generated using an inverter operated in the dimming mode. Input voltage, current, and frequency were about 0.68 kV, 77 mA, and 83 kHz, respectively. Ambient air was used as a feeding gas. The distance between the plasma device end and the mushroom sample was 1 cm. PTW was produced by treating 50 mL of deionized (DI) water with plasma at 5 cm for 30 min.

### 2.3. Measurement of pH and Level of NO_x_ and H_2_O_2_ in Plasma-Treated Water

The pH of PTW (plasma-treated water) and DI (deionized) water (control) was measured using a portable pH meter (Oakton Instruments, Vernon Hills, IL, USA). Concentrations of NO_x_ and H_2_O_2_ in PTW and DI water were measured using a QuantiChrom^TM^ Nitric Oxide Assay Kit (BioAssay Systems, Hayward, CA, USA) and an Amplex™ Red Hydrogen Peroxide/Peroxidase Assay Kit (Molecular Probes, Eugene, OR, USA) following the manufacturer’s instructions. Three replicate measurements were performed in each PTW and control sample, and the measurements were repeated three times.

### 2.4. Treatment of Mushrooms and Antimicrobial Activity Assessment

Ten mushroom pieces were treated with NTAPPJ or PTW. For treatment with NTAPPJ, mushroom pieces placed in a 35 mm Petri dish were exposed to the plasma jet at 1 cm for 10, 15, and 20 min. For treatment with PTW, mushroom pieces were placed in 10 mL of PTW and incubated for 10, 15, and 20 min. After treatment with NTAPPJ or PTW, mushroom pieces were inoculated with bacteria by placing them into 1 mL of an *E. coli* suspension for 10 min and then submerging them into DI water for 10 s to wash off the non-adherent bacteria. After drying for 30 s at room temperature (25 °C) to remove the excess DI water, the washed mushroom pieces were blended mechanically with 0.5 mL saline using a pestle, and subsequently placed in a micro-tube. The extract of the mushroom was then serially diluted with saline. The diluted extract (100 μL) was spread onto TSA plates and incubated at 37 °C; the CFU number was counted after 16 h. CFU number was counted from three replicate plates in each experiment, and the experiment was repeated three and four times for PTW and NTAPPJ, respectively.

### 2.5. Live and Dead Assay of Adherent Bacteria on Mushroom Surface

To analyze the viability of adherent bacteria, mushroom pieces were treated with NTAPPJ or PTW for 15 min and then inoculated with *E. coli* as described in Section 2.4. After brief washing and drying, adherent bacteria on the mushroom surface were stained using the LIVE/DEAD^®^ BacLight^TM^ bacterial viability kit (Molecular Probes, Eugene, OR, USA) according to the manufacturer’s protocol. Equal volumes of SYTO9 dye and propidium iodide (PI) were mixed thoroughly. Three microliters of the mixture of dye solution were added into 1 mL of saline containing mushroom pieces. After 15 min of incubation at 25 °C in the dark, the stained samples were observed using a fluorescent microscope (Nikon, Tokyo, Japan) at 480_ex_/500_em_ nm (for the SYTO9 stain) and 490_ex_/635_em_ nm (for the PI stain). Bacteria with intact cell membranes (live) were stained fluorescent green (SYTO9 stain), whereas bacteria with damaged membranes (dead) were stained fluorescent red (PI stain). Pictures of stained bacteria were taken. The experiment was repeated three times.

### 2.6. Scanning Electron Microscopy

The ultrastructure of the mushroom surface and bacteria attached after plasma treatment was analyzed using a scanning electron microscope (SEM) (JEOL, Tokyo, Japan). Mushroom pieces were treated with NTAPPJ or PTW for 15 min and then inoculated with *E. coli,* as described in Section 2.4. Plasma-treated mushrooms with or without bacterial inoculation were prepared for SEM analysis as described previously [33]. Mushroom samples were incubated in Karnovsky’s fixative (2% (*v*/*v*) paraformaldehyde and 2% (*v*/*v*) glutaraldehyde in 1 × PBS) at 4 °C overnight. After washing the fixed mushroom sample with PBS three times, a secondary fixation was conducted by incubating the sample in 1% (*v*/*v*) osmium tetroxide for 2 h at room temperature in the dark. Mushroom samples were washed twice with PBS and then dehydrated by serial incubation in 30, 50, 70, 80, 90, and 100% ethanol (twice). Dehydrated samples were dried via incubation in hexamethyldisilazane (HMDS) twice for 15 min and then mounted on carbon tape. After being coated with platinum, samples were examined using SEM.

### 2.7. Lipid Peroxidation Assay

The colorimetric lipid peroxidation (MDA) assay kit (Abcam, Cambridge, UK) was used to assess the peroxidation of bacterial membrane lipids. Mushroom pieces were treated with NTAPPJ or PTW and then inoculated with *E. coli,* as described in Section 2.4. After a brief washing and drying, mushroom pieces were placed in a conical tube and incubated in 2 mL saline with shaking for 10 min to detach the bacteria from the mushroom surface. After a brief spinning of the tube, saline containing detached bacteria was transferred into a new tube. The detached bacterial suspension (100 µL) was treated with lysis solution, and then TBA (thiobarbituric acid) reagent was added to produce MDA (malondialdehyde; by-product of lipid peroxidation)–TBA adduct following the manufacturer’s protocol. The reaction mixture with TBA was incubated at 95 °C for 60 min and then kept in ice for 10 min to stop the reaction. The reaction mixture was centrifuged at 3000 rpm for 5 min, and the supernatant was collected. The absorbance of the supernatant was determined using a microplate reader (BioTek Instruments, Winooski, VT, USA) at 532 nm. The concentration of MDA in the sample was calculated using a standard curve, which was obtained using a known amount of MDA supplied with the assay kit. The assay was repeated two times, and three replicate measurements were performed in each assay.

### 2.8. Test for Hydrophilicity and Redox Potential of the Mushroom Surface

To examine the hydrophilicity of mushroom surfaces after treatment with NTAPPJ, mushroom water absorption was examined. Mushroom pieces were treated with NTAPPJ for 10, 15, and 20 min. The weight of the control (no plasma treatment) and treated mushroom pieces was measured before soaking in DI water for 10 min. Mushroom pieces were removed from the water, the surface was briefly dried, and the weight of the mushroom pieces was re-measured. Absorbed water was assessed by calculating the difference in weight before and after soaking in water. The assessment was performed from three replicates in each experiment, and the experiment was repeated four times.

The redox potential of the mushroom surface was assessed using water-soluble tetrazolium (WST) solution as described in a previous study [28]. WST solution (EZ-Cytox, Daeil Lab Service, Seoul, Korea) was prepared by diluting in 0.9% NaCl solution (ratio of 1:10). The optical density (OD) of WST solution (at 450 nm) was maintained at approximately 1.5 using pure magnesium powder. Two pieces of mushroom treated (15 min) with NTAPPJ and PTW, respectively, and a control piece (no treatment) were each dropped into 1 mL of WST solution, and the solution was recovered after 30 s. The OD of WST formazan, a product of WST reduction, was measured at a wavelength of 450 nm using a microplate reader (BioTek Instruments, Winooski, VT, USA). An increase or decrease in the OD value indicates the oxidizing or reducing potential of the mushroom surface, respectively. The assay was repeated six times, and three to four replicate measurements were performed in each assay.

### 2.9. Quality Evaluation of the Mushroom after Treatment: pH, Antioxidant Activity, Lipid Peroxidation, and Weight Loss

For quality evaluation of mushrooms after plasma treatment, mushrooms were treated with NTAPPJ or PTW for 15 min as described in Section 2.4. Control and treated mushrooms were stored in a refrigerator (4 °C and 70% relative humidity) for 0, 7, and 12 days. After storage, mushrooms were analyzed. The morphology of mushrooms was photographed after storage.

For pH measurement, one whole mushroom was crushed thoroughly in 1 mL of DI water using a mortar and pestle and the pH of the extract was then measured. Three replicate measurements were performed in each sample, and the experiment was repeated two times.

An MDA assay was performed using the MDA assay kit (Abcam). Treated and control mushroom samples were ground with liquid nitrogen, and the ground powder was mixed with a lysis buffer supplied with the kit. Further extraction and MDA assay were performed as described in Section 2.7. Three replicate measurements were performed in each experiment, and the experiment was repeated two times.

The total antioxidant activity of mushrooms can be evaluated by assessing the scavenging effect of DPPH (2,2-diphenyl-1-picrylhydrazyl), a biomarker of free radicals [34]. Equal amounts of treated and control mushrooms were ground with liquid nitrogen, then 1 mL of ethanol was added. After vigorous mixing, the mixture was centrifuged at maximum speed for 5 min, and the supernatant was transferred to a new tube. The extract was mixed homogeneously with the DPPH reagent solution (Sigma, St. Louis, MO, USA), and the mixture was incubated at room temperature in the dark for 30 min. The absorbance (Ab) of the mixture was measured at 517 nm using a microplate reader (BioTek Instruments). Pure ethanol instead of mushroom ethanol extract was used as a blank. Total antioxidant activity of the sample was calculated as a percentage of DPPH scavenged; % DPPH scavenged = (Ab_Blank_ − Ab_Sample_)/Ab_Blank_ × 100. Three replicate measurements were performed.

To assess the water loss of the mushroom, the mushrooms’ weight was measured before plasma treatment. After plasma treatment and storage, the weight of treated and control mushroom samples was measured again. The percentage of water loss was calculated as; % water loss = (Weight before treatment − Weight after treatment)/Weight before treatment × 100. Three replicate measurements were performed.

### 2.10. Statistical Analysis

All data were indicated as average and standard deviation of replicate measurements as stated earlier. The significance between data points was determined using the t-test and single-factor ANOVA test. Statistical analysis was performed using GraphPad Prism v 8.0 (GraphPad Software, San Diego, CA, USA) software. Significant differences were established as *p* value less than 0.05 and denoted using alphabetical letters; different letters indicated significant difference. Groups having at least one same letter were not significantly different.

## 3. Results

### 3.1. Plasma Produces Reactive Species in Air and Water

NTAPPJ (Figure 1a,b) produces various reactive and active species such as atomic oxygen and active nitrogen species as already demonstrated in a previous study [32]. After DI (deionized) water was treated with NTAPPJ for 30 min (Figure 1c), the pH of water (around 7) was significantly decreased to around 4 (Figure 1d). The concentrations of H_2_O_2_ and NO_x_ in PTW was significantly increased to around 1.6 and 2.8 μM, respectively, compared to those in non-treated water (~0.5 μM).

### 3.2. NTAPPJ and PTW Can Enhance Antimicrobial Activity on Mushroom Surfaces

After mushroom pieces were exposed to NTAPPJ and then inoculated with *E. coli*, the CFU number of *E. coli* extracted from the inoculated mushroom was significantly reduced (about 60–75%) with time-dependent exposure (Figure 2a). When mushroom pieces treated with PTW were inoculated with *E. coli*, the viability of *E. coli* extracted from the mushrooms reduced (about 35–85%) over treatment time, and a significant reduction was observed for treatments of 15 and 20 min (Figure 2b).

SEM analysis showed that the number of bacteria attached was reduced on mushroom surfaces treated with NTAPPJ, compared to that on the control mushroom surfaces (Figure 3 and Appendix A). PTW-treated mushroom surfaces showed a similar or smaller number of attached bacteria in different areas, compared to that on control surfaces (Appendix A). The greatest decrease in bacterial attachment was found on NTAPPJ-treated mushroom surfaces. The morphology of the bacteria on the mushroom surface was not significantly different between the control and NTAPPJ or PTW treatments (Figure 3). However, the size of a greater number of bacteria, attached on the surface of mushroom treated with NTAPPJ or PTW, was smaller than that on the control (Figure 3 and Appendix A).

To discover whether the attached bacteria were alive or dead, we performed the LIVE–DEAD assay for bacteria attached on treated and control mushroom surfaces. Figure 4 shows the fluorescent pictures of attached bacteria on mushroom surfaces after being stained with SYTO9 (living bacteria) and PI (dead bacteria). A greater number of bacteria were stained with SYTO9, emitting green fluorescence on the control compared to that on the NTAPPJ- or PTW-treated mushroom surfaces (Figure 4a and Appendix A). Fluorescent red bacteria stained with PI were more frequently found on NTAPPJ- or PTW-treated mushrooms than on the control mushroom surfaces (Figure 4a and Appendix A).

To test whether bacteria continued to die on the plasma-treated mushroom surface, the inoculated mushrooms were incubated. Our preliminary data showed no significant change in the proportion of live and dead bacteria after incubation for 1 and 5 min, indicating no continuous bacterial death on the NTAPPJ- and PTW-treated mushroom surfaces (Appendix A).

### 3.3. NTAPPJ- and PTW-Modified Mushroom Surfaces

To discover the mechanism(s) for bacterial inactivation on the NTAPPJ- and PTW-treated mushroom surface, the mushroom surface topography was analyzed. No significant difference in surface roughness was observed between the control and PTW-treated mushroom surfaces (Figure 5 and Appendix A). However, the surface of NTAPPJ-treated mushrooms displayed uneven roughness and clumping masses in several areas, compared to the control surface (Figure 5 and Appendix A).

Hydrophilicity was significantly increased on NTAPPJ-treated mushroom surfaces. Water absorption by mushrooms was dramatically elevated after treatment with NTAPPJ for 10–20 min (Figure 6a). The weight of the mushrooms treated with NTAPPJ for 10, 15, and 20 min was 0.23, 0.20, and 0.19 g after being soaked in water, respectively, whereas control mushrooms weighed 0.122 g (Figure 6a). This indicated that water absorption was increased in plasma-treated samples, probably caused by the increased surface hydrophilicity.

The redox potential of the mushroom surfaces was assessed using water-soluble tetrazolium (WST) solution [28]. When WST solution, partially oxidized using magnesium powder (WST + Mg) and with the optical density (at 450 nm) maintained at 1.5, was applied to the mushroom surfaces with no treatment, NTAPPJ treatment, and PTW treatment, a significant reduction in the OD value of WST + Mg solution was observed in NTAPPJ-treated mushrooms (Figure 6b). Reduction in the OD value indicated that WST was oxidized by the mushroom surfaces treated with NTAPPJ. This indicates that the mushroom surface treated with NTAPPJ had reductive potential, oxidizing the objects on the surface. The OD values of WST + Mg solution were not significantly changed after being placed on control or PTW-treated mushrooms (Figure 6b).

Because the mushroom surface had reductive potential after NTAPPJ treatment, we examined the peroxidation of the membrane lipids of bacteria inoculated on the mushroom surface by measuring the level of MDA as a by-product of lipid peroxidation. MDA production in *E. coli* inoculated on mushroom surfaces treated with NTAPPJ and PTW was significantly greater than that in those inoculated on non-treated mushroom surfaces (Figure 6c). This indicates that peroxidation of bacterial membrane lipids progressed more actively on the surfaces of mushrooms treated with NTAPPJ and PTW.

### 3.4. The Quality of the Mushroom after Plasma Treatment

The quality of mushrooms as a food was evaluated after treatment with NTAPPJ and PTW. When stored at 4 °C, all mushrooms were partly dried after 7 days and severely dried after 12 days (Figure 7). NTAPPJ-treated mushrooms dried faster than the controls (no treatment) after 7 and 12 days (Figure 7). No significant change in dryness was observed between control and PTW treatment (Figure 7). After 12 days of storage, fungal growth was noticed in all groups. However, the fungal contamination was less in both NTAPPJ- and PTW-treated groups compared to that in the control group (Figure 7).

The dryness of mushrooms is related to water loss. We measured the weight of mushrooms before and after storage to discover the effect of plasma on the water content in mushrooms. After storage, about 7% and 22% weight loss was observed in non-treated mushrooms stored at 4 °C for 7 and 12 days, respectively (Figure 8a). This may be caused by loss of water in mushrooms during storage. Water loss was greater in mushrooms treated with NTAPPJ, compared to that in the non-treated ones. There was a significant increase in mushroom water loss (about 18–35%) on the 7th and 12th day after NTAPPJ treatment (Figure 8a). However, PTW-treated mushrooms showed no significant weight loss compared to controls (non-treated mushrooms) (Figure 8a).

The MDA content in mushrooms, a measure of lipid peroxidation, increased over storage time in all groups (Figure 8b). The MDA content in non-treated mushrooms was not significantly different from that of NTAPPJ- and PTW-treated mushrooms up to 7 days of storage (Figure 8b). On the 12th day of storage, MDA content was significantly greater in mushrooms treated with NTAPPJ and PTW than in the control (Figure 8b).

Antioxidant activity of mushrooms measured using DPPH scavenging showed no significant difference between non-treated and NTAPPJ- or PTW-treated mushrooms on day 0 of storage (Figure 8c). After 7 days of storage, antioxidant activity in both NTAPPJ- and PTW-treated groups was significantly greater than in the control group (Figure 8c). However, there was no significant difference between the control and NTAPPJ- or PTW-treated mushrooms after storage for 12 days (Figure 8c).

The pH of the mushrooms was recorded over the storage period because it can be a critical factor for mushroom quality, reflecting metabolic changes [35]. As shown in Figure 8d, pH was significantly decreased in NTAPPJ- and PTW-treated mushrooms compared to that in the control on days 0 and 7. After storage for 12 days, PTW-treated mushrooms showed a pH significantly lower than non-treated ones (Figure 8d). However, all pH values were included between 6 and 7 (neutral range).

## 4. Discussion

In this study, we demonstrated that surface of Enoki mushrooms (a living organism) became antibacterial after plasma treatment. Similar results were observed in Xu et al.’s study in which bacterial and fungal contaminations in button mushrooms (*Agaricus bisporus*) were significantly reduced after mushrooms were soaked in plasma-treated water [27]. However, our study focuses on the effects of pre-treatment (priming effect) using plasma on the resistance of treated mushrooms against bacterial contamination, whereas Xu et al. demonstrate the effects of post-treatment with plasma on natural microbiota already present in mushrooms. Similar antibacterial activities of plasma or plasma-treated water and buffers have already been observed in many post-harvested fruits and vegetables [19,24,36,37].

A point of our study distinguished from previous studies is the antimicrobial effects of pre-treatment of living material using plasma (effects of priming treatment). There have been many reports demonstrating antimicrobial modification of surfaces of non-living materials by plasma [28,29,30] or surface grafting with antimicrobial compounds assisted by plasma (for review, see [31]). However, modifications of living surfaces have limited number of studies. Our study provides evidence showing that atmospheric pressure non-thermal plasma can change the surface of living organisms to be antimicrobial. In addition, bacterial attachment was reduced, and the attached bacteria died on plasma-treated mushroom surfaces in our study. This indicates that modification of mushroom surfaces caused by plasma (particularly NTAPPJ) can generate double antibacterial effects—inhibition of attachment and induction of death. Similar results were also observed on plasma-treated titanium surfaces, in which titanium surfaces treated with a plasma jet prevented bacterial attachment, and the attached bacteria died out [28].

Our results suggest several possible mechanism(s) behind bacterial inactivation on plasma-treated mushroom surfaces. In most studies, the antimicrobial role of plasma treatment is related to the generation of reactive species and the modification of surface properties of the treated sample, which can create an unfavorable environment for microbial survival [29,30]. Analysis of ultrastructure, redox potential, and hydrophilicity of mushroom surfaces indicates that there are changes in the nature of the treated mushroom surface, different from that of a non-treated one. Hydrophilic changes and increased redox potential of mushroom surfaces treated with NTAPPJ may have caused the decreased bacterial attachment and death of attached bacteria. Surface hydrophobicity enhances bacterial adhesion, which facilitates biofilm formation and survival in various environments [38,39]. Because the bacterial cell surface is itself hydrophobic, bacteria are more prone to attach to hydrophobic surfaces and less strongly to hydrophilic surfaces. Therefore, increased the hydrophilicity of mushroom surfaces after treatment with NTAPPJ may have helped to reduce bacterial adhesion. Our results also showed that a greater number of attached bacteria were dead on NTAPPJ-treated than on non-treated mushroom surfaces. The redox potential result suggested that the reductive potential of the NTAPPJ-treated mushroom surface was elevated, which might oxidate the bacterial cell attached to it. We observed that the oxidation of WST was significantly higher on the NTAPPJ-treated mushroom surface. Furthermore, a higher level of lipid peroxidation, measured as the content of MDA, was observed in bacteria attached on NTAPPJ-treated mushroom surfaces. Thus, bacteria on NTAPPJ-treated mushroom surfaces were more easily oxidized, leading to death.

On PTW-treated mushroom surfaces, the redox potential was not significantly different, and bacterial attachment did not seem significantly affected. However, lipid peroxidation was significantly increased in bacteria. ROS detected in PTW may not be enough to change the redox potential of mushroom surfaces but could produce secondary species that cause bacterial lipid peroxidation. In this study, we did not examine the production of secondary species, and further investigation might be needed. The modification of surface properties seems to be more obvious in mushrooms treated with NTAPPJ than in those with PTW treatment. However, surface modification together with the generation of secondary species can be still considered as possible factors explaining the increased level of bacterial lipid peroxidation observed on surfaces of PTW-treated mushrooms. In addition, the chemical and molecular basis of surface modification (increased hydrophilicity and redox potential) should be clarified to understand the general impact of NTAPPJ and PTW treatments on the cellular structure of mushrooms (post-harvest products) in future studies.

One issue in post-harvest sanitation and sterilization is the effect of plasm treatment on the food quality of fresh produce. Regarding the post-harvest quality of fresh produce treated with plasma, the outer appearances such as discoloration [24,40,41], firmness [42], and photosynthetic efficiency [43] were analyzed in previous studies. In this study, we tried to examine the quality by analyzing factors related to both appearance and metabolic activity. The most prominent difference after plasma treatment was the water content in mushrooms. Significantly higher water loss was observed in NTAPPJ-treated mushrooms than in the control and PTW-treated mushrooms after storage. This suggests that PTW treatment may be a good option for mushrooms sanitation in terms of maintaining food quality. Antioxidant activity was significantly increased after 7 days of storage, after NTAPPJ or PTW treatment; this may explain no significant effect on lipid peroxidation. The pH reduction in NTAPPJ- and PTW-treated mushrooms during storage indicates a more active metabolism in these mushrooms because active metabolic processes decrease cellular pH. Generally, pH level was maintained within 6–7 (around neutral range), and therefore food quality did not seem to be damaged. Our study also demonstrates that storing mushrooms as a fresh food for more than one week is not recommended. Plasma treatment does not worsen the mushroom’s quality within this period, except for its water content. However, after 12 days storage, dryness and other properties were worsened after plasma treatment, and this may be partly due to the aging effect. Interestingly, microbial (maybe fungal) growth was less severe in plasma-treated mushrooms during storage. This suggests that a trade-off between antimicrobial properties and the food quality of mushrooms should be considered when applying plasma treatments for mushroom sanitation.

## 5. Conclusions

This study shows the effects of direct NTAPPJ treatment and indirect PTW treatment on mushroom sanitation. Both treatments changed mushroom surfaces to be more antibacterial with similar efficiency. However, the mechanisms of antibacterial action were different between NTAPPJ and PTW treatments; with increased oxidizing ability and inhibition of bacterial attachment in NTAPPJ, but not in PTW. Here, direct treatment using jet plasma showed control of microbial contamination. Nonetheless, considering the food quality parameters, PTW treatment was more acceptable. For scale-up treatment, the treatment condition and timing should be optimized in further studies.

## Figures and Tables

**Figure 1 foods-10-01888-f001:**
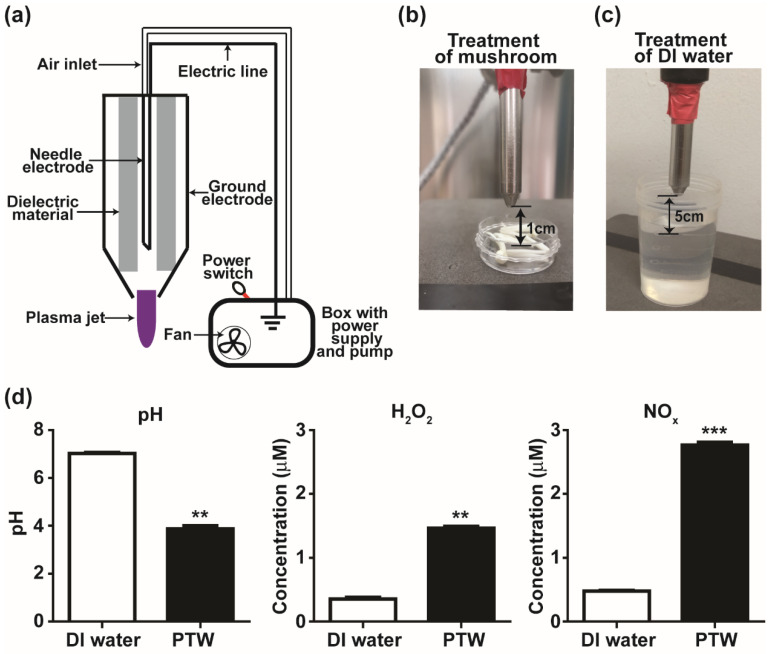
Treatment with NTAPPJ (non-thermal atmospheric pressure plasma jet) and properties of PTW (plasma-treated water). (**a**) Schematic view of NTAPPJ device. (**b**) Photo of treatment of mushroom pieces with NTAPPJ. (**c**) Photo of treatment of deionized water with NTAPPJ. (**d**) pH of PTW and level of H_2_O_2_ and NO_x_ measured in PTW. Each value is the average of nine replicate measurements. ** *p <* 0.01, *** *p <* 0.001.

**Figure 2 foods-10-01888-f002:**
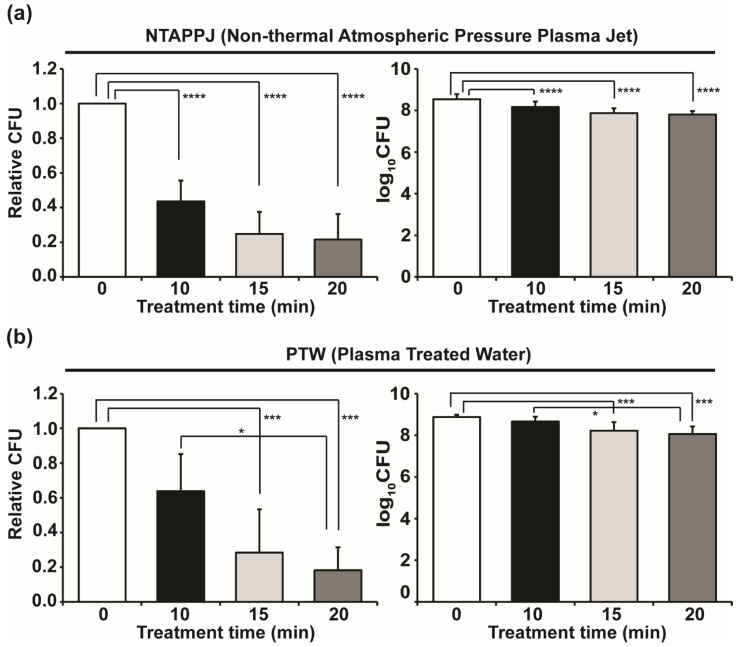
Viability of *E. coli* extracted from inoculated mushroom after treated with NTAPPJ (**a**) and PTW (**b**). Each value is the average of 9 (PTW) and 12 (NTAPPJ) replicate measurements. Only significantly different groups were indicated; * *p <* 0.05, *** *p <* 0.001, **** *p <* 0.0001.

**Figure 3 foods-10-01888-f003:**
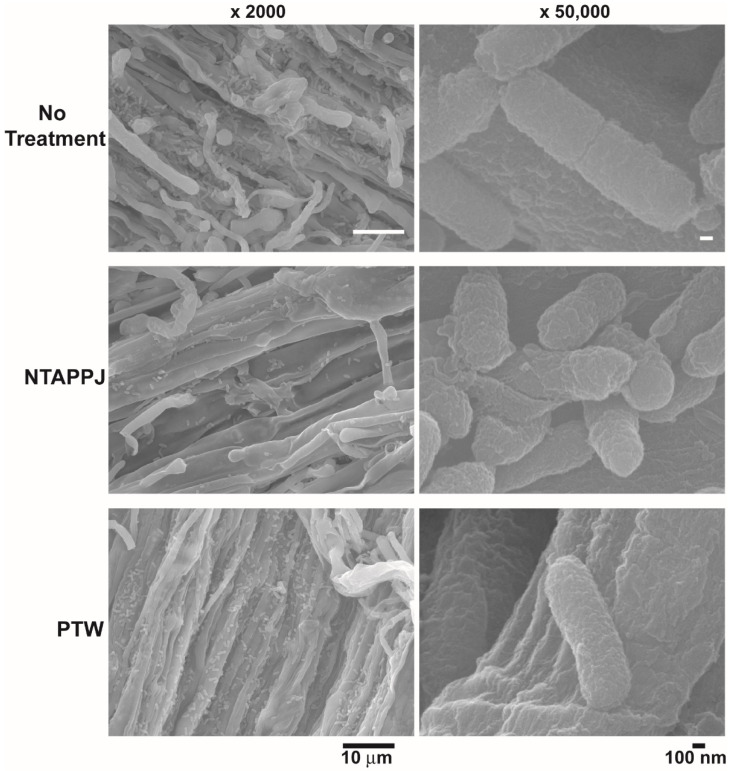
SEM images of the surface of mushroom treated with NTAPPJ and PTW after inoculation with *E. coli*.

**Figure 4 foods-10-01888-f004:**
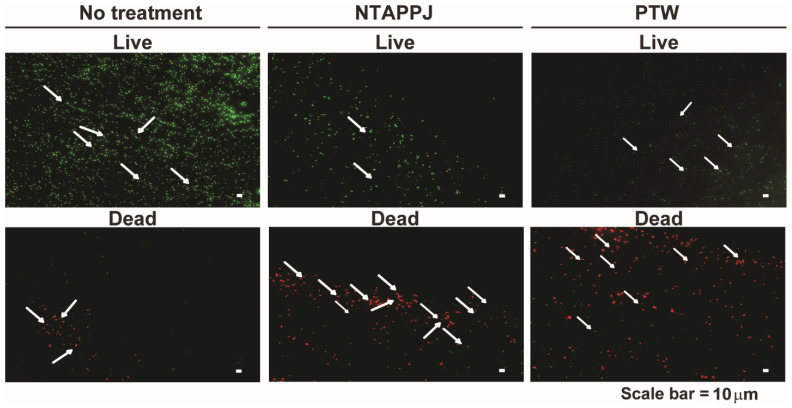
Live and dead bacteria on mushroom surfaces. Images of live (green fluorescence) and dead (red fluorescence) bacteria on mushroom surfaces with no treatment, NTAPPJ, and PTW. Treatment time was 15 min. Arrows indicate bacteria.

**Figure 5 foods-10-01888-f005:**
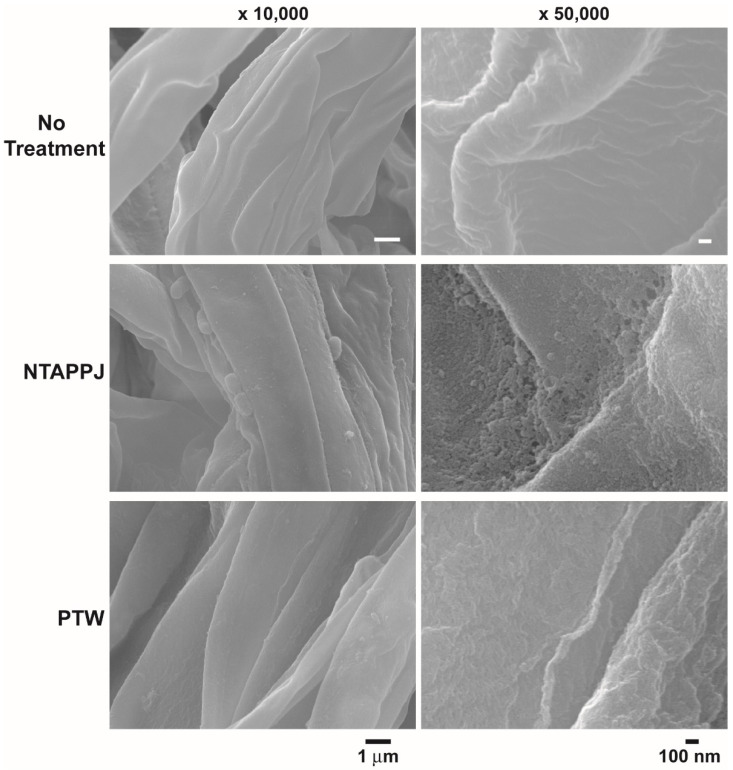
Ultrastructure of mushroom surface with no treatment, NTAPPJ treatment, and PTW treatment.

**Figure 6 foods-10-01888-f006:**
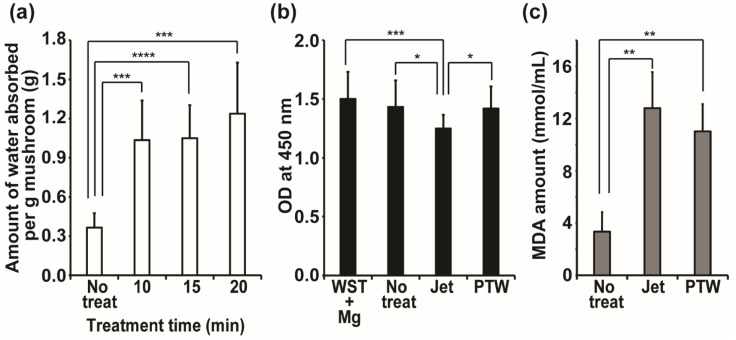
Hydrophilicity (**a**), redox potential (**b**), and lipid peroxidation of bacteria (**c**) on the surfaces of mushrooms. Values were the average of 12, 19, and 6 measurements in (**a**), (**b**), and (**c**), respectively. Only significantly different groups were indicated; * *p <* 0.05, ** *p <* 0.01, *** *p <* 0.001, **** *p <* 0.0001.

**Figure 7 foods-10-01888-f007:**
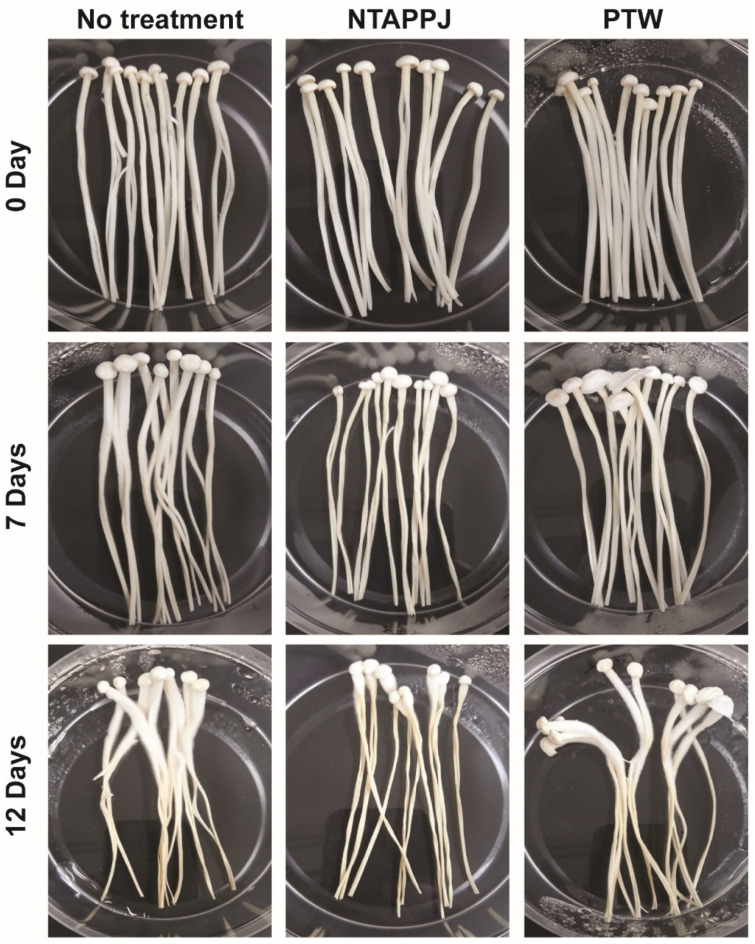
Appearance of *F. velutipes* mushrooms stored at 4 °C for 7 and 12 days.

**Figure 8 foods-10-01888-f008:**
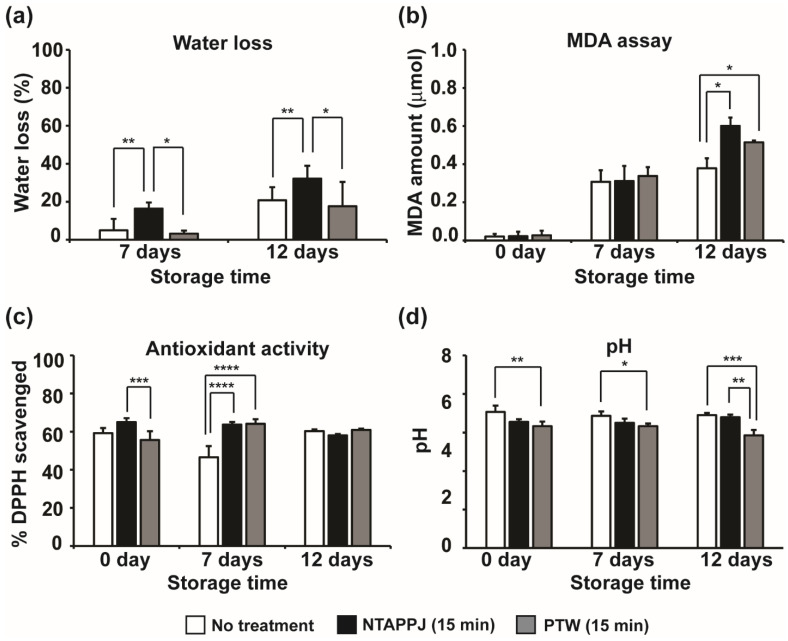
Measurement of water loss (**a**), lipid peroxidation (**b**), antioxidant activity (**c**), and pH (**d**) in mushroom after no treatment, NTAPPJ treatment, and PTW treatment. Values were the average of three to six measurements. Only significantly different groups were indicated; * *p <* 0.05, ** *p <* 0.01, *** *p <* 0.001, **** *p <* 0.0001.

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
