# Peer review of "Effects of Pre-Treatment Using Plasma on the Antibacterial Activity of Mushroom Surfaces"

_foods, 2021, doi:10.3390/foods10081888_

Round 1

Reviewer 1 Report

This study consists of analyzing the response of the previous treatment with non-thermal atmospheric pressure plasma on the improvement of the microbial conditions in mushroom (although only of E.coli).

The title should be more precise in relation to the study being presented.

Within the abstract, the last sentence of the paragraph (which corresponds to the conclusions) should be more specific.

Material and methods section. The methodology is well described, however, it is not detailed precisely how many times each analysis has been carried out. In the statistical analysis section, it says “three or more replicate”. As befits a scientific study, it should be more accurate. In addition, it is not indicated if the process has been carried out again with different samples to verify that the results are robust.

The Student's t test is indicated when there are only two groups of analysis. However, when there are more than two groups, as is the case with some of the results presented, the recommended test is an ANOVA. In addition, the form of presentation of the statistics in the figures is not adequate. Nor is it indicated with which statistical package it was made. I recommend that a statistician be consulted for proper analysis and proper presentation of results.

Regarding the discussion of the results, a good explanation of the results obtained is made, but the comparison with the results found in other papers is very scarce. The list of bibliographic references is scarce. In addition, it is recommended to cite more recent research studies.

Author Response

Reviewer 1

This study consists of analyzing the response of the previous treatment with non-thermal atmospheric pressure plasma on the improvement of the microbial conditions in mushroom (although only of E. coli).

The title should be more precise in relation to the study being presented.

Response) As the reviewer suggested, we changed the title to “Effects of pre-treatment using plasma on the antibacterial activity of mushroom surfaces”.

Within the abstract, the last sentence of the paragraph (which corresponds to the conclusions) should be more specific.

Response) We thank the reviewer for the valuable comment. We extensively modified the abstract for clarity.

Material and methods section. The methodology is well described, however, it is not detailed precisely how many times each analysis has been carried out. In the statistical analysis section, it says “three or more replicate”. As befits a scientific study, it should be more accurate. In addition, it is not indicated if the process has been carried out again with different samples to verify that the results are robust.

Response) We indicated the number of replicates and experimental repetition in Materials and Methods section. We also indicated the total number of replicates used in the analysis in figure legends. In addition, we revised some area in methods for providing more accurate information.

Revised area: lines 116-117, 129-131, 144-145, 175-176, 184-185, 195-196, 205-207, 222-223, 228, figure 1 and 2 legends

The Student's t test is indicated when there are only two groups of analysis. However, when there are more than two groups, as is the case with some of the results presented, the recommended test is an ANOVA. In addition, the form of presentation of the statistics in the figures is not adequate. Nor is it indicated with which statistical package it was made. I recommend that a statistician be consulted for proper analysis and proper presentation of results.

Response) Thanks to the reviewer for pointing out this. We apologize sincerely for our mistake.  In this study, t-test has been applied for analysis consisting of two groups and ANOVA has been applied for more than two groups. The statistical program used in this study is Graphpad prism. In graphs, significance only between control and each treatment was indicated as * (p < 0.05), ** (p < 0.01), and *** (p < 0.001). We revised section 2.10 in Materials and Methods.

Revised area: line 230-235

Regarding the discussion of the results, a good explanation of the results obtained is made, but the comparison with the results found in other papers is very scarce. The list of bibliographic references is scarce. In addition, it is recommended to cite more recent research studies.

Response) Thanks to the reviewer for suggestion. We revised the discussion section addressing comparisons of the obtained data with prior studies.

Revised area: lines 370-392, 418-433

Reviewer 2 Report

The manuscript titled “Non-thermal atmospheric pressure plasma can enhance the antimicrobial quality of mushroom surfaces” demonstrated the benefits of pre-treatment of mushroom surfaces with non-thermal atmospheric pressure plasma jet (NTAPPJ) or plasma-treated water (PTW). The experimental design and obtained data successfully demonstrate the potential of this approach to prevent food spoilage. However, the background information presented on post-harvest antimicrobial technologies, post, and pre-treatment of contaminated food, their advantages, and disadvantages, are lacking. Due to this lack of depth, the novelty of the work becomes questionable. Therefore, the authors must rewrite this section to reflect better the state of this research field and the challenges that the authors were aiming to address with this experiment.

Additional comments are presented below:

  1. Lines 14-15 and lines 20-21 of the abstract are unclear and need to be rewritten.
  2. The source of the E. coli bacteria has not been mentioned in the materials and methods section. The authors must mention what microbial collection the bacteria was purchased or obtained from. Similarly, the authors are encouraged to provide more details on the source of the mushrooms.
  3. Page 3, Lines 99 and 107: “After gentle drying…” The authors are advised to provide more details on the drying process, as this step can greatly influence the bacterial attachment results.
  4. The authors are advised to elaborate more on the impact of the plasma treatments on the structure of mushrooms. The findings discussed on Page 13, lines 369-384, do not offer substantial information regarding the possible impacts of the plasma treatments on the cellular structure of the mushrooms. Furthermore, this amendment can be followed by the applicability of the plasma treatment to preserve other harvested produce, such as herbs.
  5. The manuscript suffers from poor grammar, and the authors are advised to format the manuscript thoroughly. For example, on Page 2, lines 58-59: “Antimicrobial effects of pre-treatment of foods with plasma are rarely explored.” The sentence implies that the pre-treatment of food with plasma for antimicrobial purposes has infect been previously attempted. If so, this section needs to be clarified since the novelty of this work becomes ambiguous due to the language used here. Another example would be the title of section 3.3, “3.3. NTAPPJ and PTW modify mushroom” which should be changed to 3.3. NTAPPJ and PTW modified or treated mushrooms. This comment also applies to the previous comment about the lack of clarity in the abstract.

Author Response

Reviewer 2

The manuscript titled “Non-thermal atmospheric pressure plasma can enhance the antimicrobial quality of mushroom surfaces” demonstrated the benefits of pre-treatment of mushroom surfaces with non-thermal atmospheric pressure plasma jet (NTAPPJ) or plasma-treated water (PTW). The experimental design and obtained data successfully demonstrate the potential of this approach to prevent food spoilage.

However, the background information presented on post-harvest antimicrobial technologies, post, and pre-treatment of contaminated food, their advantages, and disadvantages, are lacking. Due to this lack of depth, the novelty of the work becomes questionable. Therefore, the authors must rewrite this section to reflect better the state of this research field and the challenges that the authors were aiming to address with this experiment.

Response) Thanks to the reviewer for suggestion. We revised the introduction section adding the background details, the significance of studies and the lacking points of prior studies. We also added references.

Revised area: lines 55-82, references

Additional comments are presented below:

Lines 14-15 and lines 20-21 of the abstract are unclear and need to be rewritten.

Response) As the reviewer suggested, we extensively modified the abstract for clarity and specificity.

Revised area: lines 10-24

The source of the E. coli bacteria has not been mentioned in the materials and methods section. The authors must mention what microbial collection the bacteria was purchased or obtained from.

Response) We added information about E. coli strain in the Materials and Methods section.

E. coli K12 (KCTC1116) was graciously provided by Korean Collection for Type Cultures in Korea Research Institute of Bioscience & Biotechnology (Jeongeup-si, Jeollabuk-do, Korea) and used to inoculate the mushroom surfaces.”

Revised area: lines 88-91

Similarly, the authors are encouraged to provide more details on the source of the mushrooms.

Response) We added more details on the source of mushrooms in the Materials and Methods section.

“Enoki mushrooms (Flammulina velutipes) were purchased as vacuum sealed package from a local market (Seoul, Korea) for this study and visually checked to avoid damage and contamination. The purchased mushroom was identified by expert personnel.”

Revised area: lines 85-87

Page 3, Lines 99 and 107: “After gentle drying…” The authors are advised to provide more details on the drying process, as this step can greatly influence the bacterial attachment results.

Response) We provided the details about the drying process.

“After drying for 30 seconds at room temperature (25 ℃) to remove the excess DI water, the washed mushroom pieces were blended mechanically with 0.5 mL saline using a pestle, and subsequently placed in a micro-tube.”

Revised area: lines 125-127

The authors are advised to elaborate more on the impact of the plasma treatments on the structure of mushrooms. The findings discussed on Page 13, lines 369-384, do not offer substantial information regarding the possible impacts of the plasma treatments on the cellular structure of the mushrooms. Furthermore, this amendment can be followed by the applicability of the plasma treatment to preserve other harvested produce, such as herbs.

Response) We thank the reviewer for pointing out this important matter. According to the findings from our study, plasma treatment is likely to modify the surface structure of mushroom because mushroom surfaces become more hydrophilic and show the increase in redox potential after plasma treatment (more specifically NTAPPJ treatment). Surface modification may be also possible in PTW treated mushrooms because the increased level of bacterial lipid peroxidation was observed on surfaces of PTW treated mushrooms. The chemical and molecular basis of surface modification (increased hydrophilicity and redox potential) was not analyzed in this study. However, those should be clarified to understand the general impact of NTAPPJ and PTW treatments on cellular structure of mushrooms (post-harvest products) in future study. We added sentences in discussion section addressing this.

Another analysis was performed on food quality of plasma treated mushrooms such as the outer appearances and metabolism related properties which were also frequently examined in previous studies. We also added sentences in discussion section addressing this.

Revised area: lines 419-432

The manuscript suffers from poor grammar, and the authors are advised to format the manuscript thoroughly.

Response) We thoroughly rechecked the grammar of sentences in text.

For example, on Page 2, lines 58-59: “Antimicrobial effects of pre-treatment of foods with plasma are rarely explored.” The sentence implies that the pre-treatment of food with plasma for antimicrobial purposes has infect been previously attempted. If so, this section needs to be clarified since the novelty of this work becomes ambiguous due to the language used here.

Response) We thank the reviewer for pointing out this issue. Our focuses in this study were 1) sanitation of a new post-harvested fresh produce, mushrooms, which were not frequently studied, rather than fruits and vegetables and 2) sanitation effect of plasma pre-treatment on living products, in other words, effect of priming treatment using plasma on sanitation of fresh produce.  Many studies have showed the sanitation of post-harvested fresh produce harboring natural microbiota or artificially inoculated microorganisms. However, sanitation effects after pre-treatment of fresh produce with plasma and then the inoculation on it with microorganisms have been rarely studied.  This was actively studied using non-living materials, but not frequently using living materials like fresh produce. Our study focuses on living materials. We extensively revised introduction section for clarification and added more references.

Revised area: lines 55-82, references

 Another example would be the title of section 3.3, “3.3. NTAPPJ and PTW modify mushroom” which should be changed to 3.3. NTAPPJ and PTW modified or treated mushrooms.

Response) As the reviewer suggested, the title of the section 3.3 has been changed to “NTAPPJ and PTW modified mushroom surfaces”

This comment also applies to the previous comment about the lack of clarity in the abstract.

Response) We extensively revised the abstract section as the reviewer suggested.

Revised area: lines 10-24

Round 2

Reviewer 1 Report

The paper has been remarkably improved, after the corrections made by the authors.

Regarding the statistical analysis, it is not only necessary to present the significance of the statistical differences between the different groups studied. It is also necessary to present which groups are different from others through a post-hoc test (indicating it with different letters). And in this way reflect it in the text where the results obtained are indicated. Review the presentation of the statistics in Figures 2, 6 and 8. I insist on the need to consult a statistician. 

Author Response

The paper has been remarkably improved, after the corrections made by the authors.

Regarding the statistical analysis, it is not only necessary to present the significance of the statistical differences between the different groups studied. It is also necessary to present which groups are different from others through a post-hoc test (indicating it with different letters). And in this way reflect it in the text where the results obtained are indicated. Review the presentation of the statistics in Figures 2, 6 and 8. I insist on the need to consult a statistician. 

Response) We thank the reviewer for helpful comments. We agree with the reviewer’s opinion. We indicated significance differences in figure 2, 6, and 8 using letters and added sentences explaining this in section 2.10 and figure legends.

Revised area: lines 233-236, figure 2, 6, and 8, lines 260-262, 314-315, 355-357

Reviewer 2 Report

Authors did a good job to revise the manuscript.

Author Response

Authors did a good job to revise the manuscript.

Response) We thank the reviewer so much. We also rechecked the spell in the text as the reviewer suggested.